# Food Addiction and Psychosocial Adversity: Biological Embedding, Contextual Factors, and Public Health Implications

**DOI:** 10.3390/nu12113521

**Published:** 2020-11-16

**Authors:** David A. Wiss, Nicole Avena, Mark Gold

**Affiliations:** 1Fielding School of Public Health, University of California Los Angeles, Los Angeles, CA 90095, USA; dwiss@ucla.edu; 2Department of Neuroscience, Icahn School of Medicine at Mount Sinai, New York, NY 10029, USA; nicoleavena@gmail.com; 3Department of Psychology, Princeton University, Princeton, NJ 08540, USA; 4School of Medicine, Washington University in St. Louis, St. Louis, MO 63130, USA

**Keywords:** food addiction, eating disorder, obesity, stress, trauma, early life adversity, adverse childhood experience, dopamine, epigenetics, biopsychosocial

## Abstract

The role of stress, trauma, and adversity particularly early in life has been identified as a contributing factor in both drug and food addictions. While links between traumatic stress and substance use disorders are well documented, the pathways to food addiction and obesity are less established. This review focuses on psychosocial and neurobiological factors that may increase risk for addiction-like behaviors and ultimately increase BMI over the lifespan. Early childhood and adolescent adversity can induce long-lasting alterations in the glucocorticoid and dopamine systems that lead to increased addiction vulnerability later in life. Allostatic load, the hypothalamic-pituitary-adrenal axis, and emerging data on epigenetics in the context of biological embedding are highlighted. A conceptual model for food addiction is proposed, which integrates data on the biological embedding of adversity as well as upstream psychological, social, and environmental factors. Dietary restraint as a feature of disordered eating is discussed as an important contextual factor related to food addiction. Discussion of various public health and policy considerations are based on the concept that improved knowledge of biopsychosocial mechanisms contributing to food addiction may decrease stigma associated with obesity and disordered eating behavior.

## 1. Background 

The quest to discover the precise mechanisms of hedonic overeating began decades ago. While many theories have been proposed, none have been widely accepted, and the obesity epidemic continues to grow. The *Nutrition Transition* theory describes a global trend toward consumption of processed foods that are low in fiber and high in added sugars and fats [1]. The changing global food landscape in the past four decades have increased access to convenient “snack” foods and decreased time spent preparing foods at home [2]. Several lines of research have explored the idea that highly palatable foods can alter brain reward pathways. For example, a landmark study showed that dopamine (DA) receptors were significantly lower in individuals with obesity [3]. Soon after, investigators documented overlapping neuroimaging characteristics in humans with obesity and those with substance use disorders (SUDs), showing reductions in DA-D2 receptors [4]. It was then suggested that individuals may overeat to compensate for DA-D2 receptor dysfunction [5]. To date, it is not clear whether these neurochemical associations are a cause of addiction-like overeating or a consequence [6]. However, similar to other addictions, changes that occur in obesity show that food reinforcement adapts, strongly implicating biological underpinnings. Given the limited success in reversing the obesity trends, a better understanding of the various biopsychosocial mechanisms may help inform public health efforts. 

Bart Hoebel pioneered the concept of food addiction (FA) research using animal models, showing evidence of bingeing, withdrawal, craving, and concomitant changes in dopaminergic and opioidergic systems in response to overeating sugar [7,8,9,10,11,12,13,14]. In rodent studies, early life adversity (ELA) has been shown to induce alterations in DA neuronal activity and synaptic function [15], impacting reward-directed behavior and partially accounting for individual variation along the mesolimbic DA projection [16]. More recently it has been shown that chronic stress dysregulates the reward system, promotes addiction-like eating, and contributes to the development of obesity [17]. Furthermore, palatable diets buffer against the negative impact of social stressors in juvenile rats [18]. Interestingly, environmental enrichment (larger space with conspecifics and novel objects) reduced sugar seeking and consumption [19]. Other rodent studies documented early and persistent alterations in amygdala circuitry and function following exposure to ELA, which were not diminished when the stressor was removed [20]. This suggests that ELA is not always redeemable by subsequent experience. At present, there is a gap in our understanding of how various forms of stress, trauma, and adversity link to addiction-like eating in real-world settings, particularly when viewed in social context, as well as over the lifespan. 

In humans, various forms of ELA are associated with illicit drug use later in life [21,22,23]. In addition, there are established links between ELA and obesity [21,24], however, the exact mechanisms are not understood. A recent systematic review on childhood obesity implicated stress as a midstream factor that can lead to “junk food” self-medication and subtle addiction, in order to alleviate uncomfortable emotional states [25]. In a nationally representative sample of young adults (n = 10,813) exposure to multiple types of child maltreatment predicted excessive sugary beverage consumption [26]. In a Brazilian sample (n = 7639) FA was independently associated with early life physical and sexual abuse [27]. A positron emission tomography (PET) study also found that long-term exposure to adversity is associated with reduced striatal DA synthesis capacity [28]. Functional magnetic resonance imaging (fMRI) studies have linked ELA to blunted subjective responses to reward-predicting cues [29], and to altered connectivity in the extended reward network, leading to increased vulnerability to FA and obesity later in life [30]. While there are sufficient data that describes life course associations between ELA and adult weight outcomes [24], the actual biological mechanisms are less understood, which is a primary focus of this review. Another aim is to integrate psychologically relevant contextual factors such as weight stigma and pathological dieting. 

The purpose of this review is to focus on literature from FA as well as obesity in the context of exposure to trauma, stress, and adversity, in an effort to answer three questions: (1) is FA a biologically plausible explanation for a life course association between ELA and obesity? (2) how might other relevant psychological, social, and environmental factors contribute to FA and to obesity? (see Figure 1) and finally, (3) what does it mean for public health? For simplicity, we have conceptually merged stress/trauma/adversity (STA) at several points throughout, particularly when reviewed outside of the context of early life, however we acknowledge these are not identical concepts. We also acknowledge that FA does not always lead to obesity, and that obesity can occur in the absence of FA. Additionally, ELA is used synonymously with adverse childhood experience (ACE) to describe exposures in the first 18 years of life. This review draws from literature across multiple disciplines in order to consider both individual and population health perspectives, and to describe contextual factors related to the neurobiology of FA. It is important to translate obesity science into a relevant social context, in order to identify achievable intervention targets which may have a meaningful impact upstream. 

### 1.1. The Biopsychosocial Model & Other Foundational Theories

Social and biological processes overlap and are inextricably linked. However, research methods are often incapable of capturing all features of an observed phenomenon, such as the various drivers of obesity (see Figure 1). Another example is how addiction neuroscience overlooks key social factors such as exclusion and marginalization which would make these findings more clinically relevant [31]. Biopsychosocial models were originally proposed by Engel as a new way to understand health and disease, by considering influence from various domains [32]. Biopsychosocial obesity research has found that lower educational attainment is associated with higher BMI, after adjusting for biological (energy intake and expenditure), psychological (decisional balance) and social (support) factors [33]. A biopsychosocial approach to childhood obesity should consider the (1) biology of the child (2) family environment and immediate psychosocial influences and (3) wider environmental, social, and cultural influences [34]. This creates opportunity for collaboration across multiple academic and clinically-focused disciplines. The current review employs a biopsychosocial perspective on FA, considering obesity as one possible outcome. This manuscript also incorporates Krieger’s *Ecosocial Theory* which emphasizes the social production of disease over biomedical individualism, describing “embodiment” as the biological incorporation of social and ecological circumstances into everyday life [35]. A *Life Course Perspective* is used to link ELA to adult health [36,37,38]. Finally, a *Developmental Psychology* perspective views human development as relational, pertaining to dynamics (e.g., community features) which require individuals to be contextually situated into multidirectional and reciprocating ecological systems [39,40,41]. 

### 1.2. Food Addiction & Eating Disorders 

With the validation of the *Yale Food Addiction Scale* (YFAS) in 2009 [42] and the updated YFAS 2.0 in 2016 [43], FA in humans has been operationalized across hundreds of studies. At the present time, FA has not been recognized as an official eating disorder (ED) in the Diagnostic and Statistical Manual (DSM) of Mental Disorders. Unique aspects of addictions include the importance of the substance, withdrawal, and tolerance, whereas unique aspects of EDs include restraint/rules, and shape/weight concerns [44]. It is well-established that dietary restraint/restriction can lead to rebound bingeing [45] yet it remains unclear if this is a cause or consequence of FA symptoms (discussed in Section 5.1). Thus, disordered eating characterized by dietary restraint provides important context for FA data. It has been recently suggested that the presence of dieting behavior must be carefully evaluated in order to separate the FA “signal” from the “noise” [46]. For example, ED research has identified significant overlap between FA and bulimia nervosa (BN), with FA symptoms improving when BN symptoms remit [47]. FA prevalence is the highest in BN [48] compared with other EDs, suggesting that FA treatment models should consider symptom contribution from dietary restraint and other compensatory behaviors. It has been proposed that FA is a transdiagnostic disorder associated with neurobiological vulnerability in certain people, who are more susceptible to using food as a coping mechanism [49]. It has also been shown that FA predicts a worse treatment outcome in patients with binge eating disorder (BED) [50].

Among those with an ED diagnosis, Brewerton (2017) proposed that the presence of FA be conceptualized as a meaningful correlate of post-traumatic stress disorder (PTSD) severity and symptoms [51]. For example, data from the Nurses’ Health Study II has shown that severe physical and sexual abuse are associated with a 90% increase in FA risk [52]. The same dataset also suggested that symptoms of PTSD are associated with an increased prevalence of FA [53]. In a sample of 301 overweight and obese women, the association between FA and childhood trauma remained significant after adjusting for potential confounders such as socioeconomic status (SES) [54]. In a sample of bariatric surgery seeking patients (n = 1586), elevated ACE scores correlated with an increased likelihood of screening positive for FA [55]. A recent meta-analysis showed that multiple ACEs increased the odds of adult obesity by 46% (95% CI: 28–64%) [24] but several unmeasured confounders likely influence this estimate, such as the presence of EDs and SUDs, which frequently cluster, co-occur, and lead to weight fluctuations [56,57]. Therefore, risk estimates between childhood adversity and adult obesity would likely be higher after adjusting for these diagnoses often associated with dietary restriction and weight control, however this has not been formally tested. While data linking ELA to EDs are robust, only recently has it been shown that FA symptoms can mediate this pathway, as well as exacerbate ED symptoms significantly across all forms of childhood maltreatment [58]. Although EDs are not directly featured in Figure 1, the constructs of dietary restraint and weight stigma are used to contextualize important associations between EDs and FA. Notwithstanding, there are likely paths from ELA to obesity that are better captured by more classic ED pathology (e.g., BED) rather than FA, which are not directly featured by the model. 

## 2. Food Addiction Neuroscience & Social Context

A frequent criticism of FA data in clinical settings is that the measure itself does not account for restrained eating [46] (discussed in Section 5.1). Another criticism is that it remains unclear how to intervene once FA has been detected. A recent systematic review of mostly pilot and feasibility studies concluded that currently there are no empirically supported psychosocial interventions for FA [59]. The authors recommend that clinicians assess for comorbid ED, and if present, provide evidence-based treatments (e.g., cognitive behavioral therapy) for those conditions. There is growing support for the FA construct in studies using the YFAS as well as neuroimaging studies on obesity that do not use “food addiction” terminology, with several examples provided below. It is worth noting that many authors reject the FA term in favor of other language such as eating addiction [60], or with additional qualifiers such as refined or processed food addiction [61,62], and even food use disorder [63]. Figure 1 proposes that FA is one driver of obesity, although there are several others, including some not captured by the model (discussed further in Section 2.1). 

Neurobiological overlaps between obesity and addiction have been described within the mesolimbic pathway between the ventral tegmental area (VTA) and the ventral striatum, with further projection to limbic (amygdala and hippocampus) and cortical regions (prefrontal cortex [PFC] and cingulate gyrus) [64]. Recent data suggests that among 110 healthy lean adults, exposure to a Western-style diet for one week led to rapid declines in hippocampal-dependent learning and memory, as well as appetitive control [65]. Research has shown that obesity (similar to SUD) is associated with deficits in executive functioning, an umbrella term encompassing the higher-order cognitive processes that help people take goal-directed action [66]. In a sample of women with obesity (n = 36), FA severity has been associated with impaired decision-making, compared to controls [67]. Resting-state fMRI data has shown decreased functional connectivity in the frontal gyrus in adults with obesity (n = 20) compared to controls [68]. A large cross-sectional study of children ages 9–11 (n = 2700) showed that increased BMI is associated with a reduced mean cortical thickness as well as lower executive functioning [69]. A follow-up report from the same study (n = 3190) suggested that BMI is associated with PFC development as well as diminished working memory [70]. Interestingly, a nationally representative sample of US adults (n = 4769 mean age 29) found that obesity is associated with poor working memory in women, but not men [71]. While tempting to consider that biological sex differences explain these findings, social context would suggest that the experience of weight stigma (discussed in Section 5.2), which is higher in women than men [72] may be a contributing factor. Recent data on school-age children (n = 176) suggests that weight-related stereotype threat (fear of confirming a negative stereotype) may explain working memory deficits more so than excess body weight [73]. 

Among patients with obesity (n = 224), FA is more closely correlated with psychological factors (depressive symptoms, quality of life) than with metabolic parameters (BMI, fat percentage, waist circumference) [74]. In a small sample of adult community members (n = 52), individuals with FA had significantly higher scores on depressive symptoms, emotion dysregulation, emotional eating, demand characteristics, motives, impulsivity, and family history of mental health problems and addiction [75]. Impulsivity can be defined as decision-making with limited forethought (rash-spontaneous behavior), having strong associations with FA [76,77,78,79]. Impulsivity hinders inhibitory control and is associated with increased intake of food [80] and drugs [81], often heightened in response to novel stimuli [82]. Delay discounting (preference for “smaller sooner” rather than “larger later” rewards) is closely associated with impulsivity and has been correlated with YFAS scores [83]. These authors believe it to be a predisposing factor rather than a consequence, although bidirectionality is likely. It has been suggested that impulsivity-related domains such as lower self-control, higher reward sensitivity, and negative affect help explain some similarities between addiction and obesity [84]. While impulsivity has heritable components linked to the mesocorticolimbic system [85] as well as serotonin-related candidate genes (e.g., HTR2A) [86], the potential for these traits to be influenced by epigenetic modification following psychosocial adversity will be explored in Section 3.1. 

### 2.1. Food Environment as a Driver of Food Addiction & Obesity 

The proposed pathways in Figure 1 suggest that FA may partially mediate the relationship between the food environment and obesity. Other pathways also exist. For example, census tract data have been used to show that wealthier neighborhoods (using median income) have better access (physical availability) to markets with healthier foods compared to poor neighborhoods [87]. Considering the importance of the built environment, neighborhood features (e.g., crime) that discourage outdoor physical activity are consistently linked to higher BMIs [88]. Recognizing that built, socioeconomic, and social characteristics co-occur [89], several investigators have advocated for a better understanding of theory-driven mediators and moderators in the relationship between neighborhood context and obesity [90,91]. Given the link between STA and FA, unsafe environments associated with lower SES neighborhoods are likely to impact BMI through increases in reward-based eating. It has been established that diet quality tends to follow a SES gradient [92]. It has also been shown that parental fruit/vegetable consumption is linked to adolescent fruit/vegetable consumption [93], suggesting that the home food environment is important. Not eating dinner as a family has also been linked with increased BMI in kindergarten age children, regardless of SES [94]. It has been proposed that the maltreatment-obesity association is spurious, driven by confounding through the home food environment. However, after testing, researchers have found limited confounding influence [95] which supports arguments in favor of biological mechanisms. 

Innovative methods that assess food environments include examining the (1) ratio of fast-food to full-service restaurants (2) ratio of bars/pubs to liquor stores and (3) presence of markets [96]. Multilevel models typically adjust for individual factors (education, hours of walking per week) as well as neighborhood factors (deprivation, walkability score). Perhaps the combination of food environment features matters more than individual components [96]. While several studies have described neighborhood “food swamps” (high density of high-calorie junk food) as predictive of obesity [97], few studies have looked at the potential roles of psychosocial pathways (mental health and wellbeing) [98]. A recent systematic review found that overall psychological resources (i.e., stress) had more consistent evidence of mediation than external neighborhood in the relationship between SES and BMI [91]. One study (n = 1112 adults) showed that paths from neighborhood characteristics to BMI could be partially explained by psychological distress and measures of inflammation [99] (discussed in Section 3). Taken together, FA is one potential pathway linking the food environment to obesity, however there is a “backdoor path” [100] through SES to STA, as well as a pathway that may not include FA (i.e., through the built environment). Comprehensive biopsychosocial frameworks cannot be tested or explained by any single study. There is conceptual support for the theory that the external environment (quick, cheap, highly palatable foods) is an upstream driver of EDs however this not been thoroughly investigated (discussed in Section 6.1). Figure 1 represents a synthesis of literature reviewed so far, as well as a roadmap for subsequent sections. 

### 2.2. Socioeconomic Status 

Given the inverse relationship between SES and BMI [101,102], obesity can also be viewed as a social phenomenon. This negative relationship has been shown in numerous countries outside of the US (e.g., Netherlands, Turkey, Morocco, South Asia) [103]. Hot-spot analysis in the US shows that higher BMI clusters are more likely in socioeconomically disadvantaged minority neighborhoods [104]. Meanwhile, large datasets (n = 43,864) have shown that obesity risk is decreased when positive contextual factors (maternal mental health, school safety, and child resilience) are present [105]. A large population-based cohort from the UK (n = 18,733) found that home-based deprivation was more closely associated with changes in child BMI than school-based deprivation [106]. On the other hand, some authors believe that the root cause of ACEs are largely based in the community, originating from an accumulation of contextual risk factors beyond a child’s control, including family history, failed attachment, safety/security, and neighborhood risks [107]. While models of addiction and obesity are incomplete without psychosocial context, current research methods cannot adequately contextualize risk. Hence, literature from multiple disciplines was used to synthesize our conceptual model, which could be expanded with additional constructs such as resilience. 

To illustrate further, a recent study showed that individuals with higher ACE scores were more likely to report not finishing high school, unemployment, and living below the poverty level [108]. Sustained activation and loss of capacity to respond to chronic stress might lead to a higher risk of illness and disease among people in lower SES categories [109]. It has been recognized that the processes which mediate the relationship between ELA and adult obesity might differ between men and women [110,111,112]. For example, a recent systematic review found that perceived stress from structural racism and weight stigma among black women creates negative emotions which predict emotional eating [113]. These stressors may then increase metabolic disturbance. Obesity itself can be a stressful state due to high prevalence of weight stigma [114] (discussed in Section 5.2), as highlighted by the feedback loop in Figure 1. Obesity may be driving changes in stress biology rather than stress biology driving obesity [115]. Next, we consider the impact of ELA (as well as STA more broadly) from a life course perspective, describing precise (as well as candidate) mechanisms by which social factors impact health, seen primarily through recent SUD and obesity research. 

## 3. Biological Embedding of Stress, Trauma, & Adversity 

The fetal and infant origins of adult disease was proposed by Barker in 1990 [116]. This focus on the biological basis of disease gave rise to concepts such as allostatic load (AL), defined as the “cost of chronic exposure to fluctuating or heightened neural or neuroendocrine response resulting from repeated or chronic environmental challenge” [117]. This can be operationalized using a range of biomarkers that indicate inflammation and long-term “weathering.” A simpler definition of AL is the price of adaption that leads to disease states over time. It has been suggested that frequent activation of the stress response and the failure to shut off allostatic activity creates “wear and tear” [118]. A landmark study showed that higher AL scores were associated with poorer cognitive and physical functioning, increasing the risk of cardiovascular diseases independent of sociodemographic risk factors [119]. Higher levels of AL (indexed by measures of blood pressure, C-reactive protein, fibrinogen, cholesterol ratio, triglycerides, and cortisol) have been observed in higher weight individuals [120]. This research suggested that these cumulatively elevated biomarkers link to decreased inhibitory control, highlighting the potential for disordered eating to become very difficult to overcome, similar to drug addiction. 

It has been suggested that inflammatory mediators act on cortico-amygdala (threat) and cortico-basal ganglia (reward) circuits in a manner which predisposes individuals to “self-medicating” behaviors such as drug use, smoking, and the excess consumption of highly palatable foods [121]. Such behaviors further propagate inflammation and create a self-sustaining feedback loop. The “neuroimmune network hypothesis” proposes that ELA amplifies the communication between the brain and the immune system, promoting low grade peripheral inflammation [122]. The “glucocorticoid cascade hypothesis” posits that stress hormones impair brain function which further increases cortisol levels [123]. Adolescents exposed to childhood adversity have larger pituitary gland volume, associated with lower cortisol awakening response [124]. These authors propose that attenuation of hypothalamic-pituitary-adrenal (HPA) axis function may derive from stress-induced chronic hyperactivation during childhood. Heightened susceptibility may be due to differences in corticotrophin-releasing hormone (CRH) within the HPA axis, responsible for the output of cortisol [125]. Individual differences in inflammatory reactivity might explain why people have differing susceptibility to the consequences of stress, which may include neuroinflammation from stress-induced pro-inflammatory cytokines [126]. A recent cohort study of nine- and ten-year-old children showed that pro-inflammatory diets (i.e., high in saturated fats) increase neuroinflammation in reward-related brain regions, which in turn lead to further unhealthy eating and obesity [127].

A meta-analysis of 1781 people documented significantly decreased hippocampal volumes following ELA, with weaker evidence of increased amygdala volumes [128]. Alterations in corticolimbic circuitry following exposure to trauma make adolescents (n = 64) less able to relax and more vulnerable to risky behavior [129]. Other observable neural changes following ELA include (1) structural variation in gray and white matter (2) functional variation in brain activity and functional connectivity and (3) altered neurotransmitter metabolism [130]. In line with what has been observed in rodent studies [131], these effects might not be restricted to one’s own lifespan but may also be transmitted to offspring [132]. It is well established that paternal drug exposure has long-lasting consequences including altered drug sensitivity in subsequent generations [133] but only recently has intergenerational transmission of trauma consistent with epigenetic explanations been described [134]. In animal models, the effects of maternal care on developing DA pathways and reward-directed behavior may account for individual differences in the mesolimbic DA system [16]. In social context, addictions may promote compromised parenting increasing the possibility that suboptimal care may be provided to the next generation. There is a timely need for longitudinal studies that capture the precise biological mechanisms that link ELA to addictions over time, as well as their consequences. 

### 3.1. Epigenetic Mechanisms of Biological Embedding 

Epigenetics is the study of how the environment regulates the genome, best described as changes in gene function without changes in gene sequence. DNA methylation is an enzymatically-catalyzed modification of DNA and is one plausible mechanism through which early life exposures (low SES, nutritional patterns) become biologically embedded [135]. Methylation changes are apparent even years after the exposure but can be reversible in some cases. Epigenetic processes may be a key mediator between social environments during childhood and disease risk throughout life. Both DNA methylation and demethylation mechanisms are likely recruited during early life unfavorable experiences [136]. Other forms of epigenetic modification include histone modification and noncoding ribonucleic acids [137].

A milestone study confirmed what had been previously shown in animal models: human parental care impacts epigenetic regulation of hippocampal glucocorticoid receptor gene (NR3C1) expression [138]. Such epigenetic marks that persist into adulthood may influence vulnerability for psychopathology through its impact on HPA axis function. However, it is difficult to determine if DNA methylation changes are the immediate results of ELA or a consequence of the phenotypes associated with such adversity [139]. Notwithstanding, NR3C1 has been linked to prenatal stress [140] and is the most studied gene to date related to abuse and neglect [141]. Other genes involved in HPA axis regulation such as corticotrophin-releasing factor (CRF) have been investigated [142]. It is not implausible that developmental programming of the HPA axis and subsequent regulation of the stress response might impact addiction susceptibility, thereby increasing intake of substances known to activate reward pathways, including highly palatable food. 

Epigenetic control of the expression of opioid receptor genes (mu-, delta-, and kappa-) has been reviewed in the context of SUDs [143]. While methylation at the mu-opioid receptor (MOR) gene is most strongly associated with drug addiction [144] as well as incentive motivation for processed food [145,146], decreased methylation at the kappa-opioid receptor (KOR) in the anterior insula has been shown in child abuse [147] (discussed further in Section 4.1). In this study of postmortem brain structures, the investigators were unable to detect a change in MOR expression, suggesting different epigenetic signatures associated with addictions and ELA. It is worth noting that different drugs have impacts at different brain regions and many include histone modifications in the nucleus accumbens (NAc) [148]. Other potentially relevant epigenetic modifications include the serotonin transporters [149,150,151] and proopiomelanocortin (POMC) [152]. Additional research is needed to determine how various epigenetic modifications associate with various forms of disordered eating, including FA. 

In a sample of 206 women with bulimic symptomatology, there was evidence of increased methylation of the DA-D2 gene promoter, compared to controls [153]. Taq1A polymorphisms at the D2 receptor has been well-studied and known to influence impulsive behavior [154]. Recent data shows that DNA methylation in obesity-related genes may relate to obesity risk in adolescents [155]. Increased obesity susceptibility genes (e.g., FTO) have been found in the insula and substantia nigra (brain regions involved in addiction and reward) [156]. It has been shown that the methylation status on DA signaling genes (SLC18A1 and SLC6A3) might underlie epigenetic mechanisms contributing to carbohydrate and calorie consumption, as well as fat deposition [157]. Recently authors have linked specific dietary components with the gut microbiome in an effort to determine epigenetic factors on offspring susceptibility to obesity [158]. Expression levels of candidate genes implicated in glucose and energy homeostasis (e.g., HDAC7 and IGF2BP2) could be epigenetically regulated by gut bacterial populations [159]. The link between epigenetic marks and gut microbes appear to be mediated by host-microbial metabolites acting as substrates and cofactors for key epigenetic enzymes in the host [160]. More research linking epigenetics to the microbiome is timely and warranted, particularly in the context of dysfunctional eating behavior (including both under- and overeating). 

## 4. Stress & Obesity 

While a PTSD diagnosis is associated with an altered stress response, chronic stress can exist in the absence of PTSD, and has been the focus of several investigations related to eating behavior. Multiple pathways have been described which link stress to obesity, including (1) interference with cognitive processes (executive function, self-regulation) (2) behavior (eating, physical activity, sleep) (3) physiological changes (HPA axis, reward processing, gut microbiome) and (4) production of biochemical hormones and peptides (leptin, ghrelin, neuropeptide Y) [114]. At a basic level, stress may lead to food consumption in the absence of hunger. It is established that poor executive functioning is associated with consumption of palatable food, leading to inflammation and metabolic changes promoting weight gain [161,162,163]. Other pathways which have been identified include the autonomic nervous system (cardiovascular functioning), the epigenome (intergenerational transmission), and the metabolome (profile of metabolites in body) [164]. The vagus nerve (part of the autonomic nervous system) has been identified as an important physiological stress pathway linked to gut microbiota [165]. With rising interest in the gut-brain axis, novel pathways which include FA are being explored [166]. FA can be considered as a partial mediator in the stress-obesity pathway, likely resulting from one or many of the biologically embedded pathways described herein.

To illustrate further, individual differences in neural response to food cues under stress have been observed in human neuroimaging studies [167], lending support to differential susceptibility. It is well established that amygdala function is moderated by stress-induced glucocorticoid (GC) release [20], and a less efficient HPA axis negative feedback loop may represent a deficiency in emotion and stress regulation [168]. Highly palatable foods stimulate stress hormones that alter the limbic system (emotions) and striatal (motivational) pathways, promoting further food craving and excessive intake [169]. Rewarding foods upregulate CRF in the amygdala and related limbic striatal pathways. The most direct physiological pathway is dominated by cortisol, which stimulates fat storage and changes dietary behavior through increased reward sensitivity (DA and opioid systems) and increased appetite (arcuate nucleus in the hypothalamus) [165]. Future research should attempt to clarify the biological embedding of chronic stress both in the absence and presence of diagnosed PTSD, specifically impacting reward-related pathways associated with consumption behavior. Additionally, more research is needed on biopsychosocial factors of resilience in the context of both FA and obesity. 

### 4.1. Stress & Addictions

Given the established links between stress and obesity, these links can be used to conceptualize relationships between stress and FA. The phenomenon of stress-induced reinstatement of drug-seeking is generalizable to other substances, including food [170]. To illustrate, the DA and GC systems are both highly involved in substance addictions, and ELA may induce long-lasting alterations in these systems. One of the most profound effects of stress is the activation of the HPA axis with release of CRF from the paraventricular nucleus. Human studies have shown stress exposure increases alcohol craving [171]. Both chronic stress and long-standing alcohol use promote PFC dysfunction [172]. Changes in CRF activity that result from chronic alcohol exposure within the extended amygdala network is thought to be key factor in withdrawal symptoms [173]. It has been proposed that repeated altered activity in the DA system and sustained activation of the CRF system leads to AL and negative emotional states [174]. The central thesis in Koob’s allostatic view of stress and addiction is that stress leads to changes in brain CRF that have a direct impact on addiction [175]. Withdrawal can produce elevated levels of GCs and increase release of CRF in the central nucleus of the amygdala [175]. 

It has also been suggested that increased CRF alters serotonin release in the brain which facilitates DA in the accumbens [176]. Prolonged exposure to stress can lead to irregular changes in GC receptor density (epigenetics) which may increase the reinforcing effects of alcohol and drugs [177,178]. Interestingly, a higher salivary cortisol level in response to stress has been associated with higher drop-out rates in treatment [179]. It has also been suggested that variability in stress-related genes may contribute to the ability of certain individuals to remain abstinent from heroin, possibly due to higher stress resilience [180]. Importantly, not only does STA increase addiction behaviors, some authors have suggested this association also exists in the opposite direction [181,182]. With illicit drugs, their procurement and use can predispose individuals to traumatic stress [183,184]. In animal models, chronic opioid pretreatment is able to robustly augment associative fear learning [185]. These changes were not observed when opioids were given after the traumatic event, and potentiation lasted beyond discontinuation of drug exposure. This concept has been thoroughly described as part of the withdrawal process in widely accepted addiction models [186,187,188]. However, more research is needed to understand how long-term exposure to highly palatable foods may alter one’s long-term response to stressful life experiences, and how this dynamic can play out in reciprocal and bidirectional ways, for example in the presence of weight stigma and dietary restraint (and cumulatively over time). 

A recent review of preclinical data suggests three mechanisms by which DA and GCs interact: (1) GCs upregulate tyrosine hydroxylase (rate-limiting enzyme in DA synthesis) (2) GCs down-regulate monoamine oxidase (enzyme responsible for DA removal) and (3) GCs are hypothesized to decrease DA uptake subsequently increasing synaptic DA [189]. Clearly stress enhances substance abuse-related effects at multiple points along the mesolimbic projection. The KOR system plays an important role in behavioral stress responses and has been implicated in stress-induced maladaptive responses [190]. While MOR activation produces euphoria, KOR is generally aversive and may contribute to negative affect states in withdrawal. According to some authors, it is possible that a stress-induced increase in KOR function promotes drug seeking by reducing DA transmission [190]. Meanwhile, reduced MOR has been observed in comorbid binge eating disorder and obesity [191] and across SUDs [144] which strongly suggest neurochemical overlap in these conditions, and which can persist despite weight loss or periods of drug abstinence. Any change in stress neurobiology is likely to influence reward. Based on observed deficits in the ventral striatum, reward responsiveness and processing may be a primary mediator of the effects of ELA [192]. Taken together, FA is a biologically plausible explanation for the life course association between ELA and obesity, however important contextual factors from the psychological domain deserve further consideration.

## 5. Psychological Correlates of Food Addiction & Obesity 

Thus far we have highlighted several social and environmental factors associated with STA and addiction-like eating. We have reviewed emerging data on the biological embedding of adversity, which may increase an individual’s susceptibility to FA, and potentially lead to obesity over time. Based on the overlap between FA and EDs as well as SUDs, we have recommended including these variables into statistical models which investigate weight outcomes. Finally, we have proposed a comprehensive conceptual framework to further contextualize these relationships by including two psychological (as well as socially constructed) correlates of FA, EDs, and obesity: dietary restraint and weight stigma.

### 5.1. Dietary Restraint 

Restrained eating is generally defined as a cognitive effort to eat less in order to lose weight [193], which has been viewed both as the problem and solution to obesity [194]. More recently it has become clear that theories of weight loss based on low-calorie dieting are failing, likely due to neurochemical, endocrine, and gastrointestinal factors which are not adequately captured by simple models of energy balance. While the concept of dietary restraint has been linked to some positive outcomes (e.g., weight management, prevention efforts) [195], it is included in our model as a risk factor for eating pathology, often associated with EDs (sometimes referred to as restriction). A classic study conducted by Ancel Keys in the 1940s examined the link between starvation and changes in human biology and behavior [196]. The study showed that significant (intentional) weight loss produced the onset of binge eating in 30% of participants (n = 36). Many of the individuals who were reduced to 50% of their baseline caloric intake for extended periods of time (months) began collecting recipes and cookbooks. The finding that caloric restriction leads to preoccupation with food has been widely cited in the ED literature. Meanwhile, it is less clear if deliberate efforts to eat differently (focusing on dietary quality rather than quantity) should be classified as pathological restraint. Extreme diets intended for health reasons which impair daily function have been described as “orthorexia nervosa” which appears to be growing problem [197]. Research linking FA recovery to orthorexia is timely and warranted. 

A dieting intervention on 121 females which included monitoring and restricting showed that monitoring increases perceived stress, while restricting increases the cortisol output [198]. Dieting is stressful, which may explain why engaging in dieting behaviors aimed at losing weight can actually have the opposite effect. Future iterations of Figure 1 may include an arrow directly from dietary restraint to STA, whereas in the current model there is only a backdoor path through weight stigma. A twin study from Finland (n = 4129) showed that dieters are prone to future weight gain independent of genetic factors [199]. A recent fMRI study showed that “successful” restrained eaters had stronger activation in the middle frontal gyrus and cerebellum (associated with executive function and inhibition) suggesting that food temptations may trigger processes of positive inhibition in some, but not others [200]. It is likely that altered neurochemistry from SUD and/or ELA/STA will impact the degree of success with dietary restraint. More research is needed on the impact of trauma on various eating behaviors, including restriction. It will prove important to better define dietary restraint in the context of FA recovery. 

There is considerable debate on how to approach FA from a nutritional standpoint, including incorporating FA data into the traditional ED landscape [201]. Meule has stated that “dietary restraint does not have to be dysfunctional as long as flexible elements are added” [202]. Based on available data linking FA and EDs (Section 1.2), it is proposed that restrained eating moderates the link between food environment and FA, as well as the link between FA and obesity (Figure 1). In other words, individuals engaging in dietary restraint are predicted to display higher levels of FA severity. Future research should examine the directionality as well as cumulative interplay of this relationship. Furthermore, it is proposed that individuals who meet criteria for FA and engage in dietary restraint may experience different effects on their weight status, depending on whether or not the restraint is successful, unsuccessful, pathological, or part of a restrictive ED. These theories need to be tested in both observational and experimental studies in an effort to better develop the emerging field of behavioral health nutrition. Recently, an 8-step process has been proposed to help clinicians discern FA from dietary restraint in order to inform inclusive vs. exclusive nutrition strategies [46]. The key discerning factors include the presence of SUD, PTSD, and ELA, which, if all present, can provide more confidence in the strength of an FA signal, particularly in the absence of dieting behaviors. 

### 5.2. Weight Stigma 

Weight stigma has been described as a “vicious cycle” where weight stigma begets weight gain [203]. Similar to dieting, the experience of stigmatization increases cortisol, which may drive food consumption by sensitizing the reward system [203]. In addition to increased cortisol, weight stigma also increases oxidative stress [204] providing further evidence of biological embedding. In a large sample of adolescents (n = 115,180), perceiving one’s body as overweight increases risk of suicidality [205]. Conceptually, weight stigma is similar to other forms of STA, which we consider as midstream drivers of eating behavior and subsequent weight outcomes, both through biological and psychosocial pathways. While unproven, it is possible that body dissatisfaction and self-stigma drives avoidance behaviors (e.g., weight loss to avoid adiposity) which is similar to the avoidance experience in PTSD. This may be one reason why efforts to lose weight can be persistent (or even relentless) for so many, despite the fact that weight loss efforts have been unsuccessful (or unsustainable) in the past. 

In a large national sample (n = 5129), weight discrimination was associated with overeating (specifically convenience foods) and less regular meal timing [206]. Individuals who are the target of weight stigma have been shown to decrease self-control and perceived capacity for weight management [207]. In a large sample of adolescents (n = 1497), FA and psychological distress mediated the association between weight-related self-stigma and binge eating [208]. It is worth acknowledging differences between externalized (others) and internalized (self) weight bias. It has also been shown that some people will experience longer term distress from weight stigma than others [209]. Perceptions and/or experiences of weight bias in primary care settings have been shown to negatively influence patient engagement with health care services [210]. In summary, weight stigma has emerged as an important component of obesity context, with strong arguments in favor of adopting weight-inclusive health policy [211]. While the FA explanation for weight control has been shown to decrease weight stigma among groups [212,213] it has been suggested that FA can increase internalized weight stigma among individuals [214]. More research is needed on the role of weight stigma driving dietary restraint, both as a cause and consequence of addiction-like eating. 

## 6. What Does It Mean for Public Health?

Research on biological programming attempts to identify the most critical and sensitive periods that underlie the developmental origins of later childhood and adult disease [215]. It has been suggested that the timing of adversity explains more variability in DNA methylation than the accumulation or recency of exposure [216]. It has also been observed that different dimensions of adversity have distinct influences on neurodevelopment [217]. The exact mechanisms which link human DNA methylation with psychological disorders have not been elucidated [151]. What we do know is that the cumulative effects of STA can impact neural function with significant implications for substance-seeking behaviors. All addictions share a common neurobiology and have known relationships to STA in both directions. Sugar, salt, and fat added to foods make them more palatable and reinforce “drug-like” behavior with loss of control, continued use despite consequences, binge episodes, and other similarities with traditional drugs of abuse. It is clear that obesity causes changes in opioid and DA signaling which alter reward processing [218]. Given the established links between ELA and the propensity for behavioral health disorders including SUD, ED, and FA, prevention efforts might have a meaningful impact upstream. Addressing clusters of disorders with shared underpinnings jointly may be more fruitful than a one-disorder-at-a-time approach [122].

A recent study of New Orleans children showed that neighborhood stress exerts a direct influence on obesity, after adjusting for diet and activity [219]. Such findings support the need to improve social conditions rather than efforts to address obesity at the individual level. It will be important to identify positive contextual factors such as neighborhood and school safety, as well as resilience [105] and develop community-based programs that promote these protective factors. Resiliency-building programs that reduce delay discounting may decrease addictive behaviors [220]. Meanwhile, socioeconomic differences in the quality of early life create “cumulative disadvantage” that contribute to gradients in health status [37]. SES indicators are upstream determinants of health while biological factors are more proximate determinants [221]. Neighborhood disadvantage creates social context which may become biologically embedded [222]. The impact of low SES can become embedded into inflammatory processes, the HPA axis, and neural function/structure, all of which are epigenetically controlled [135]. It is not unreasonable to assume that normalizing/improving HPA axis function may be beneficial in the treatment and relapse of addiction-related disorders. Given that epigenetic patterns are sculpted during early life [137], reducing stressors appears crucial to the long-term management of FA.

A “systems thinking” multilevel approach will be critical to reverse obesity trends. For example, trauma-informed treatment and stress management curriculum should be made available in underserved communities, starting with schools [107]. Mobilizing cross-sector interdisciplinary partnerships to connect ELA to later life health outcomes will be critical. It has been stated that “fostering increased societal awareness about toxic stress exposures that are often hidden, stigmatized, and attached to shame needs to occur across generations” [223]. Greater awareness of the biological mechanisms discussed herein are likely to reduce weight stigma, which is a known barrier for individual help-seeking behaviors in those with obesity [224] as well as SUDs [225]. Feelings of rejection associated with weight stigma and disordered eating are additional stressors which may further perpetuate a negative cycle [114,168]. A recent study showed that the FA model explanation for obesity resulted in lower stigma than the traditional “diet and exercise” explanation that attributes obesity to personal responsibility [213]. Given that weight stigma is a psychosocial contributor to maladaptive eating behavior, interventions targeting stigma (at the individual and societal levels) are warranted. 

### 6.1. Food Policy 

This paper has reviewed evidence to suggest that improving the early childhood environment might impact obesity risk and therefore should be a public health priority. Meanwhile, if reward-related neuroadaptations associated with addiction persist over time, addressing only the underlying factors may fail to create lasting changes in eating behavior, suggesting that policies targeting the food environment will also be important. Given that the food environment in the US promotes easy access to foods with addictive potential [226] it is not unreasonable to hypothesize that highly palatable foods leave a biological imprint which may perpetuate FA symptoms across the lifespan and into subsequent generations, as has been shown in animal models [131]. Western-style diets rapidly impair appetitive control, compared to those on their habitual diet [65]. Combined with heightened susceptibility to STA stemming from ELA, efforts to address the obesity epidemic may be futile without strategic multilevel interventions targeting corporate responsibility (i.e., “Big Food”) [227]. 

There is mounting evidence of the harmful effects of processed foods in contemporary diets. A recent trial comparing the caloric intake of those on ultra-processed foods (containing minimal whole foods) compared to unprocessed/whole foods for two weeks found ad libitum intake was increased by approximately 500 kcal/day on the ultra-processed diet [228]. Not surprisingly, people gained weight on the ultra-processed diet and lost weight on the unprocessed. Cross-sectional data (NHANES 2005–2014) has shown that higher consumption of ultra-processed foods is associated with excess weight and is more pronounced in females [229]. A study from Spain showed that four or more servings per day of ultra-processed foods is associated with a 62% increased hazard for all-cause mortality, where each additional serving increased all-cause mortality by 18% [230]. It remains unclear if the negative health effects are due to the direct impact of the processed foods, or the displacement of nutrient-dense high-fiber foods protective against oxidative stress and associated inflammation.

Public health interventions to increase access to healthy foods in lower SES communities have been unsuccessful in reducing obesity, therefore new approaches are needed. Identifying certain foods to be addictive may encourage collective efforts to avoid them [231] and is associated with support for policies to curb their use [232] similar to how public health officials addressed Big Tobacco. Only recently have researchers and policy makers begun to explore targeting the food environment in universal ED prevention efforts [233]. It has been suggested that “processed food addiction is the result of an intentional epidemic of addiction not an incidental by-product of Western environments” [62]. The term “processed food addiction” implicates the food industry rather than the individual. There is a critical need for increased awareness of FA and the role played by multinational food corporations in promoting processed foods with addictive qualities [214]. Evidence suggests that aggressive marketing of these foods to children, adolescents, and young adults disproportionately affects vulnerable groups [234,235,236,237,238]. While it is highly unlikely that food companies will re-formulate their products based on self-regulation, it also unrealistic to expect food-addicted individuals to regularly avoid food-related temptations. Policy support should include warning labels, industry reductions on sugar, and product bans (e.g., energy drinks) [232] while legal tools include advertising restrictions and class-action litigation [239]. Several authors have recommended policies restricting fast food advertising to adolescents [240,241]. Based on growing evidence for FA, this may be indicated. 

## 7. Conclusions 

The biological underpinnings of addictions strongly imply a role for ELA in the development of FA and obesity. Importantly, ELA can alter the physiological response to various forms of psychosocial STA across multiple body systems, which can have a cumulative impact on health behaviors over the lifespan. FA research which began in animal models has since been described in human neuroimaging studies which capture neurobiological and behavioral overlap between FA and SUDs. A biopsychosocial perspective on FA considers biomarkers such as inflammatory markers and other measures of AL, the HPA axis including the output of cortisol, epigenetic mechanisms including those that influence the HPA axis, and various structural, functional, and morphological brain changes, following exposure to ELA and STA. In order to contextualize risk, a biopsychosocial model considers the upstream drivers and fundamental causes of health disparities, such as SES and environmental (e.g., neighborhood) factors that impact food access and food choices. Furthermore, obesity frameworks should incorporate weight stigma as an important cause and consequence of the epidemic, suggested herein as a form of STA that can also become biological embedded. Finally, the role of dietary restraint has been included as an important psychological factor that should be accounted for when conceptualizing FA and obesity, particularly given the strong relationship between ELA and EDs, as recently reviewed elsewhere [46]. 

Stress proliferates over the life course and across generations, widening health disparities between advantaged and disadvantaged groups [242]. This might explain why public health nutrition interventions in low SES communities have had limited success. Consumption of highly palatable foods to “self-medicate” the long-term biological impact of chronic stress may be a critical factor in understanding the obesity crisis. This is particularly true for marginalized groups with less access (e.g., affordability) to unprocessed foods. Higher SES groups are more likely to have success in reducing addiction-like eating compared to lower SES groups who are constrained by access and resources. Public health interventions should account for the growing inequities in health outcomes. Biopsychosocial approaches that consider the cumulative interplay between social and biological factors are helpful when conceptualizing multiple systems driving substance-related disorders, whether it be alcohol, drugs, nicotine, or food [243]. A biopsychosocial model may contribute to conceptual and methodological advances in our understanding and treatment of obesity. Meanwhile, separating constructs into biological, psychological, and social factors (as in Figure 1) can be contraindicated by ecological models that emphasize the dynamic reciprocity between these levels. However, our conceptual model has discerned between these factors in order to encourage further contextual analysis of FA. 

Based on the biological plausibility of FA as a consequence of psychosocial STA, potential solutions to the obesity epidemic may include: (1) improve social conditions in order to reduce exposure to ELA, as well develop community-based programs for early intervention (2) decrease weight stigma based on FA data implying that body weight is not simply a “choice” (3) mind-body approaches (e.g., yoga, meditation) designed to improve the human stress response and (4) policy proposals aimed at the food industry to reduce exposure to highly palatable foods. More information is needed about the role of nutrition in the reversibility of unfavorable gene expression. More research is needed to investigate whether long-term dietary changes such as abstaining from highly palatable foods is even feasible. If so, will this improve the microbiome and stimulate/reverse epigenetic change and/or lead to altered reward pathways in the brain? At a minimum, it is reasonable to predict that reducing exposure to addiction-like eating can improve executive functioning. Since dietary restraint is a known risk factor for the development of EDs, drastic individual nutrition changes should be implemented in consultation with a qualified professional such as a registered dietitian nutritionist, particularly when there is underlying trauma and/or SUD. Treatment models should be trauma-informed and include staff trainings. 

FA and SUD share multiple predisposing factors including ELA which can become biologically embedded. These findings may link social determinants to specific health outcomes and elucidate pathway effects of risk across the life course. Epigenetic processes may be a key mediator between social environments during childhood and disease risk in adulthood. Mediating mechanisms such as AL, the HPA axis, DNA methylation, and altered reward sensitivity (i.e., dopamine systems) have scientific merit, however the fundamental causes of health inequalities present in society should not be overlooked. Low SES and neighborhood disadvantage remain important drivers of ELA, particularly within the context of the obesity epidemic. The cumulative effects of STA that impact neural function and heighten threat vigilance have significant implications for substance-seeking behaviors, including eating. The FA construct has gained credibility from animal and human studies reviewed herein, which may help reduce stigma associated with addiction-like behaviors, including obesity. More research is needed to understand the differential impact of inflammatory signaling markers on the brain, including assessment of blood brain barrier integrity. The study of neuroinflammation is likely to add explanatory power to our conceptual model and guide future research questions.

If the DSM accepts FA, it will lead to better treatment and eventually public health efforts to improve the national food environment and global nutrition landscape. More resources should be allocated for nutrition education during pregnancy and lactation, particularly in underserved communities where stress and adversity are high, and the food environment is suboptimal. Applying the FA framework has the potential to influence the way people view food, and to ultimately decrease addiction in future generations. FA treatment does not always require specific “food abstinence” but it does warrant reduced exposure and harm reduction strategies. Given the strong evidence that neurobiological responses to food differ among people, personalized precision nutrition interventions are warranted. In order for these strategies to be successful, cultural shifts around food norms will be necessary. Furthermore, FA is both an individual and collective health problem, and should be addressed at the societal level with broad policy interventions. We propose that unregulated promotion of addictive foods by the food industry are major contributors of obesity, particularly in the face of disadvantage and distress. Government interdiction may be required to reduce the epidemic of obesity and the growing problem of food addiction. Multidisciplinary efforts using trauma-informed integrated biopsychosocial frameworks will be necessary to reverse obesity trends. 

## Figures and Tables

**Figure 1 nutrients-12-03521-f001:**
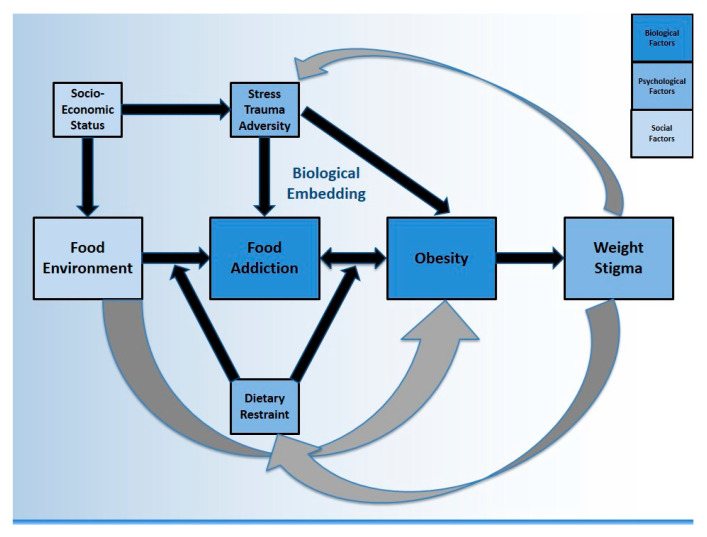
Food addiction and obesity following exposure to stress, trauma, and adversity: A biopsychosocial perspective of contextual factors.

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
