# Peer review of "Food Addiction and Psychosocial Adversity: Biological Embedding, Contextual Factors, and Public Health Implications"

_nutrients, 2020, doi:10.3390/nu12113521_

Round 1

Reviewer 1 Report

The manuscript, “Food Addiction and Early Life Adversity: Social Context and Public Health Implications” covers an important topic where there is a significant gap in the literature. A theoretically based review that highlights a roadmap for future research and public health implications is much needed. However, the current review is limited by a lack clear definitions of the constructs being covered and the organization was very hard to follow. There was a lot of review of constructs un-related to ELA with major gaps in discussion of constructs that are essential to this topic. I would suggest a significant re-organization of the manuscript, more clearly laying out the constructs being covered, and more closely focusing on ELA-related constructs (because this is the novel aspect of the current review).

Additional feedback:

  • This review includes lots of important content, but some feels tangential to the main hypothesis (and some of the information is covered in other reviews by this group, especially the content on the overlap between restraint and food addiction). I would suggest a reorganization with one overarching figure with one conceptual model and then build off of that.
  • Terms for stress/trauma/early life adversity/ACES are used interchangeably, which adds to the confusion.
  • Obesity and food addiction are used interchangeably. This is highly problematic as research finds that these constructs are only moderately correlated. This review would be strengthened by really focusing on the food addiction literature and at best using the obesity literature to provide a proof of concept (and highlight the importance of specifically investigating these topics in food addiction).
  • One of the most important factors is the ability of addictive foods to exacerbate stress responses (when being consumed or when people are withdrawing from them) and how this might impact people who have experience ELA. There is significant evidence of this in animals and emerging evidence in humans regarding withdrawal. There is also evidence that addictive foods can lead to hippocampal dysfunction that may exacerbate ELA. These seem like key topics that deserve their own sections and to be a main focus of the review.
  • Industry culpability in marketing these foods to under-resourced communities that have higher levels of stress/trauma and ELA also seems particularly relevant.
  • The central role of discussing restraint and weight stigma in this paper does not seem to fit on a paper on early life adversity and food addiction. ELA is not discussed in either of these domains and this content area has been covered in other reviews by the authors.
  • I’ve suggested a number of domains to cover that feel more relevant to the ELA thesis. There are a number of sections in the current manuscript that could be dropped or minimized to include a deeper discussion of these topics including:1) FA & Eating Disorders, 20 FA, Neuroscience & Social Context
  • Having multiple figures (including one that looks unfinished was confusing). One overarching figure would be more helpful.
  • A developmental perspective about the importance of reducing ELA in children to reduce risk for food addiction seems very important as well.
  • This article can also seem under-cited in sections. For example: “It is well established that poor executive functioning is associated with consumption of palatable food, leading to inflammation and metabolic changes promoting weight gain.” Has no citation. Similarly, “All addictions share a common neurobiology, have strong relationships to STA, and are expected to increase in the future.” This is a particularly strong claim to have without any citation and borders on out-stripping the current scientific consensus.
  • A new article was just published that may be of interest:

Bou Khalil, R.; Sleilaty, G.; Richa, S.; Seneque, M.; Iceta, S.; Rodgers, R.;
Alacreu-Crespo, A.; Maimoun, L.; Lefebvre, P.; Renard, E.; Courtet, P.;
Guillaume, S. The Impact of Retrospective Childhood Maltreatment on Eating
Disorders as Mediated by Food Addiction: A Cross-Sectional Study. Nutrients
2020, 12, 2969.

Reviewer 2 Report

The authors propose three conceptual models that describe how early life adversity (ELA) conveys risk for food addiction. These models consider biological factors, the intersection of eating and substance use disorders, and upstream psychological, social, and environmental factors. In general, the manuscript is well-written and proposes an innovative framework from which to study food addiction in future studies. My primary comment about the present manuscript concerns the delineation of symptoms of food addiction from eating disorders, and acknowledgment of the role of ELA in eating disorder development (which has an extensive literature base). The authors need to consider phenotypic overlap across these syndromes, as well as what mechanisms may distinguish these. Based on the present review it is not clear  if and how ELA conveys general or specific risk for FA/eating disorders.

While the authors briefly note ambiguity surrounding “pathological restraint,” it would be helpful if the authors expand upon the relevance of adaptive vs. maladaptive forms of restraint in the context of their model (e.g., see Schaumberg et al., 2016).

In the following sentence, the terms “predictor” or “correlate” appear to be more appropriate than “proxy”: “Brewerton (2017) proposed that the presence of FA may be used as a proxy for post-traumatic stress disorder (PTSD) severity and symptoms.”

For better clarity, please ensure that headings map onto terminology used in the proposed models.

Please revise the first paragraph of the Summary & Conclusions section to focus on the role of ELA in particular.

Reviewer 3 Report

This is a great paper, and I have very few sugggestions.  The paper is comprehensive and thought provoking, tackling some controversial issues, but with strong linkages to the empirical literature to back up the points being made.  It raises a number of good ideas for future research and policy initiatives to consider.

I am a little confused about the inclusion of “normal or underweight” in Figure 2, though, as it is not necessarily the case that the pathway from SUD would lead to “normal or underweight” before one encountered obesity.   It also seems that recovery is part of the path from “normal or underweight” to “obesity”, at least according to the narrative provided.  The narrative also provides a lot of information on the role of executive function deficits and variants of impulsivity, but these are not captured in the Figure.  Actually, the more I think this through, the more I think the figure greatly oversimplifies what is explicated in this paper, so it either needs to be more inclusive of involved variables or just deleted. 

There might just be a problem with the PDF file I reviewed, but Figure 3 seems to have some formatting problems.

Round 2

Reviewer 2 Report

The authors have been very responsive and I have no further comments.

Author Response

Dear Editors- all first-person language changes have been made.

Line 79-80 did not reference to Figure 3 but rather it is spelling out the 3 questions being answered by the paper. I have added two words to make it more clear. Finally, the formatting issues are not showing up on my end, so I am including Figure 1 both as a .ppt and .pdf separately, rather than embedded.

A Final Version along with tracked changes is included.

Thank you for the opportunity.
